# Relative contribution of diet and physical activity to increased adiposity among rural to urban migrants in India: A cross-sectional study

Sanjay Kinra[1]*, Poppy Alice Carson Mallinson[1], Jenny A. Cresswell[1], Liza J. Bowen[1], Tanica Lyngdoh[2], Dorairaj Prabhakaran[2], Kolli Srinath Reddy[2], Mario Vaz[3], Anura V. Kurpad[3], George Davey Smith[4,5], Yoav Ben-Shlomo[4], Shah Ebrahim[1]

1 Department of Non-Communicable Disease Epidemiology, London School of Hygiene & Tropical Medicine, London, United Kingdom, 2 Public Health Foundation of India, New Delhi, India, 3 St John's Research Institute, St John's National Academy of Health Sciences, Bangalore, India, 4 Population Health Sciences, University of Bristol, Bristol, United Kingdom, 5 Medical Research Council Integrative Epidemiology Unit, University of Bristol, Bristol, United Kingdom

* Sanjay.Kinra@lshtm.ac.uk

**Data Availability Statement:** Data cannot be shared publicly because of ethical restrictions. Data are available from the cohort's data management

## Abstract

### Background

In common with many other low- and middle-income countries (LMICs), rural to urban migrants in India are at increased risk of obesity, but it is unclear whether this is due to increased energy intake, reduced energy expenditure, or both. Knowing this and the relative contribution of specific dietary and physical activity behaviours to greater adiposity among urban migrants could inform policies for control of the obesity epidemic in India and other urbanising LMICs. In the Indian Migration Study, we previously found that urban migrants had greater prevalence of obesity and diabetes compared with their nonmigrant rural-dwelling siblings. In this study, we investigated the relative contribution of energy intake and expenditure and specific diet and activity behaviours to greater adiposity among urban migrants in India.

### Methods and findings

The Indian Migration Study was conducted between 2005 and 2007. Factory workers and their spouses from four cities in north, central, and south of India, together with their rural-dwelling siblings, were surveyed. Self-reported data on diet and physical activity was collected using validated questionnaires, and adiposity was estimated from thickness of skinfolds. The association of differences in dietary intake, physical activity, and adiposity between siblings was examined using multivariable linear regression. Data on 2,464 participants (median age 43 years) comprised of 1,232 sibling pairs (urban migrant and their rural-dwelling sibling) of the same sex (31% female) were analysed. Compared with the rural siblings, urban migrants had 18% greater adiposity, 12% (360 calories/day) more energy intake, and 18% (11 kilojoules/kg/day) less energy expenditure ($P < 0.001$ for all). Energy

committee (contact via http://apcaps.lshtm.ac.uk/) for researchers who meet the criteria for access to confidential data.

**Funding:** This work was funded by the Wellcome Trust, UK (https://wellcome.ac.uk/), through grant number 070797 (SE, SK). The funders had no role in study design, data collection and analysis, decision to publish, or preparation of the manuscript. George Davey Smith works in the Medical Research Council Integrative Epidemiology Unit at the University of Bristol MC_UU_00011/1.

**Competing interests:** The authors have declared that no competing interests exist. GDS is a member of the Editorial Board of PLOS Medicine. YBS is a member of the Alzheimer's Society Advisory Board and has received grant funding from the MRC, Parkinson's UK, NIHR, Versus Arthritis and Wellcome Trust.

**Abbreviations:** LMIC, low-and middle-income country; MET, metabolic equivalent task; SLI, Standard of Living Index.

intake and expenditure were independently associated with increased adiposity of urban siblings, accounting for 4% and 6.5% of adiposity difference between siblings, respectively. Difference in dietary fat/oil (10 g/day), time spent engaged in moderate or vigorous activity (69 minutes/day), and watching television (30 minutes/day) were associated with difference in adiposity between siblings, but no clear association was observed for intake of fruits and vegetables, sugary foods and sweets, cereals, animal and dairy products, and sedentary time. The limitations of this study include a cross-sectional design, systematic differences in premigration characteristics of migrants and nonmigrants, low response rate, and measurement error in estimating diet and activity from questionnaires.

## Conclusions

We found that increased energy intake and reduced energy expenditure contributed equally to greater adiposity among urban migrants in India. Policies aimed at controlling the rising prevalence of obesity in India and potentially other urbanising LMICs need to be multicomponent, target both energy intake and expenditure, and focus particularly on behaviours such as dietary fat/oil intake, time spent on watching television, and time spent engaged in moderate or vigorous intensity physical activity.

## Author summary

### Why was the study done?

- People who migrate from villages to towns and cities in India and many other developing countries have more body fat, but whether this is because of more energy intake or less expenditure or both is unclear.

- Knowing this, and the contribution of specific diet and activity behaviours, would help policy makers to develop programmes to control the rise in obesity in India and possibly other developing countries.

- In the Indian Migration Study, we previously found that urban migrants have more obesity and diabetes compared with their siblings who still live in villages.

### What did the researchers do and find?

- In the Indian Migration Study, conducted between 2005 and 2007, we collected data on urban factory workers and their spouses in four cities in India, together with their siblings who were still living in a rural area.

- Data on diet, activity, and obesity for 2,464 participants (1,232 sibling pairs of the same sex) at an average age of 43 years (30% females) were analysed.

- We found that energy intake and expenditure contributed equally to greater body fat in urban migrants; fat/oil intake, time spent watching television, and performing moderate or vigorous intensity physical activity were found to be the most important behaviours contributing to greater adiposity of urban migrants.

**What do these findings mean?**

- Energy intake and expenditure contribute equally to the increased body fat of urban migrants in India; behaviours such as fat/oil intake, time spent on watching television and performing moderate or vigorous intensity physical activity contributed most to more body fat among urban migrants.

- Programs for controlling the rise of obesity in India and possibly other developing countries should be multicomponent and target these behaviours that we have identified.

## Introduction

India has experienced a rapid increase in the prevalence of obesity due to increasing adiposity over the last two decades, which is attributed to lifestyle changes associated with urbanisation, rural to urban migration and globalisation [1–3]. Between 1991 and 2011, the urban population in India grew from 11% to 34% (380 million), and migrants from rural to urban areas accounted for around 20% of the urban population growth [4]. Migrants from rural to urban areas acquire lifestyle changes associated with urban environment, putting them at an increased risk of obesity [5,6]. Establishing the relative contribution of diet and physical activity to increased adiposity among urban migrants and the specific behavioural changes involved is vital for developing appropriate policies for control of the obesity epidemic in urbanising India and could also inform policy in other low- and middle-income countries (LMICs) with high rates of rural to urban migration [7,8].

Data on the relative contribution of changes in diet and physical activity to risk of obesity in urban migrants are lacking for India and other LMICs. Despite their utility for assessing this question, longitudinal studies on health effects of migration are almost nonexistent because of difficulties in predicting who will migrate [9,10]. A few cross-sectional studies (conducted in China, Peru, and Guatemala) have compared urban migrants to rural populations from the area of their birth; they found lower levels of physical activity and more energy intake among urban migrants, suggesting that a decline in physical activity was the primary contributor to increased adiposity among urban migrants [11–13]. However, in such comparisons, the role of unmeasured confounding due to differences in premigration characteristics of urban migrants, such as genetics, childhood home environment (including parental behaviours), and behaviours in adolescence should not be ignored [14–16]. Furthermore, these studies did not directly compare the relative contribution of diet and physical activity to greater adiposity among urban migrants.

In the Indian Migration Study conducted between 2005 and 2007, we substantially limited the role of unmeasured confounding by comparing urban migrants to their rural dwelling siblings who shared their childhood home environment (completely) and genetics (partially). The urban migrants were sampled from four factories in north, centre, and south of India (with rural siblings residing anywhere in the country) to take account of differences in lifestyles and genetics and make findings more generalisable [17]. In a previous analysis, we found that urban migrants had a higher prevalence of obesity and diabetes compared with their nonmigrant rural siblings [15]. In this study, our primary aim was to investigate the relative contribution of energy intake and expenditure to excess adiposity in urban migrants. Our

secondary aim was to identify specific dietary and physical activity behaviours contributing to this excess adiposity. Based on findings from other LMICs noted above (e.g., China, Peru, and Guatemala), we hypothesised that the greater adiposity in urban migrants was primarily due to a reduction in their energy expenditure.

## Methods

The study is reported in accordance with the STROBE guideline (S1 STROBE checklist). Detailed methodology of the Indian Migration Study has been previously reported, and the questionnaire is included as supplementary information (S1 Text) [17,18]. Briefly, the study was based at factory sites in four cities (Lucknow: Hindustan Aeronautics Ltd; Nagpur: Indorama Synthetics Ltd; Hyderabad: Bharat Heavy Electricals Ltd; and Bangalore: Hindustan Machine Tools Ltd) situated in the north, centre, and south of India. Factory workers and their co-resident spouses were recruited if they were rural-urban migrants, using employer records as the sampling frame. Each participant was asked to invite one nonmigrant sibling, preferably of the same sex and closest to them in age, still residing in their rural place of origin. Sibling pairs were examined together and all data, including measurements of skinfold thickness, were collected at the same time (i.e., cross-sectional study design). Ethical approval for the study was obtained from the institutional review board of the All India Institute of Medical Sciences. Written informed consent (witnessed thumbprint if illiterate) was obtained from the participants. The study was conducted between March 2005 and December 2007.

## Measurements

Diet was assessed by an interviewer-administered semiquantitative food frequency questionnaire [19]. The questionnaire assessed portion size and frequency of intake (i.e., daily, weekly, monthly, yearly, never) of 184 commonly consumed food items over the preceding year. A standard portion size was assigned to each food (e.g., tablespoon, ladle, bowl) by showing them to participants. A single questionnaire was used to cover the four regions of the study. Nutrient databases were developed for the study by collecting recipes from participants in rural and urban areas of each study site and using Indian food composition tables (supplemented by other international databases where necessary) to calculate nutrient content of each recipe [20–22]. Because of variation in food preparation, we used nutrient databases specific to region, urban/rural setting, and type of oil used for cooking to calculate the average daily dietary intake of energy, carbohydrate, fat, and protein. The recipes were also used to generate databases of the food group composition of each food item and used to calculate average daily food group intake. For this analysis, the following food groups were considered: fruits and vegetables (including vegetables added to preparations and salads); cereals and legumes (pulses, lentils, whole gram preparations); sugary foods and sweets (including sugar and jaggery used in preparations and beverages); meat (including meat added to mixed dishes), fish and poultry; fats and oils; and dairy (including dairy products added to preparations and beverages). The questionnaire was found to have acceptable reliability (kappa coefficients for food groups over periods ≥1 month ranging from 0.26 to 0.71) and validity against three 24-hour diet recall surveys (Spearman's rank correlation coefficients for food groups ranging from 0.25 to 0.72) [19].

An interviewer-administered questionnaire was used to assess habitual physical activity over the past month for the following domains: sleep, occupation (sitting, standing, walking, and activities more strenuous than walking at work), exercise, household, hobby, sedentary, and other (e.g., eating, dressing, travelling to and from work) [23]. Instead of a predefined list of activities, participants were asked open-ended questions to elicit a list of activities (up to 21) specific to them, e.g., 'apart from work, how do you spend your time (over last 1 month): (i)

Sports/games/exercise (e.g., walking, badminton, jogging, cricket, etc.).' The exceptions to this were for sleep, standing, sitting, walking at work, and 'other' activities that were closed questions, e.g., 'on average how many hours in a day do you sleep?' For each reported activity, additional information was gathered on its frequency and duration (in minutes per day). Physical activities were assigned metabolic equivalent task (MET) value using the Compendium of Physical Activity and WHO guidelines, supplemented by country specific values [24–26]. One MET is equivalent to 1 kcal/kg/h, corresponding to the resting metabolic rate of sitting quietly. Individual activity durations were summed to generate total daily duration of recalled activities. If this value was <24 hours, a residual time variable was generated and a standard MET value of 1.4 was applied, whereas if this value was more than 24 hours, duration of each individual activity was reduced proportional to the amount overreported [27]. For occupational activities more strenuous than walking, the Integrated Energy Index was applied to correct for unreported rests that occur in these activities [28]. Total activity was calculated as MET (h/day) by summing MET values of all activities. Time spent in categories of activity intensity was generated using previously published cut-points: sedentary (<1.5 MET); light (1.5 to 3 METs); moderate (3 to 6 METs); and vigorous (>6 METs) [29]. Physical activity energy expenditure was calculated as total activity MET (h/day) minus MET (h/day) for sleep and MET (h/day) equivalent of resting energy expenditure while being awake and multiplied by 4.183 to estimate kJ/kg/day. The questionnaire was found to have acceptable reliability (intraclass correlation coefficients over periods ≥1 month ranging from 0.26 to 0.62) and validity against physical activity assessed by uniaxial accelerometer worn ≥4 days (Spearman's rank correlation coefficient ranged from 0.18 to of 0.48 for activities of different intensities) [23].

Time lived in an urban area was estimated by collecting details about each address that the participant had lived in for longer than a year [6]. Time spent at each residence was summed (classifying the place as urban based on the Indian census definition) to estimate the total time lived in an urban area (in years). Socioeconomic position was assessed using a subset of 14 questions (out of 29) of the Standard of Living Index (SLI), which is an asset-based score use in national surveys of India. We applied the prescribed weights to generate a score ranging from 0 to 38; a higher score indicates a higher socioeconomic position [30]. Trained personnel took anthropometric measurements of height and weight from all participants during the clinic visit [17,18]. Height was measured in bare feet in the Frankfort plane, using a portable plastic stadiometer with a base plate, accurate to 1 mm (Leicester height measure, Chasmors Ltd, London). Weight was measured in light indoor clothing using a digital weighing machine with 100 g accuracy (Model PS16, Beurer, Germany). Height and weight were used to calculate the body mass index (BMI) = weight (kg) ÷ height (m)$^2$. Skinfold thickness was measured three times at the triceps and subscapular areas using Holtain calipers, and the average at each site was used to calculate percentage body fat using a standard equation for Indian population [31]. Data quality was assured through detailed protocols and 6 monthly standardisation of measurements (within a 5% margin of error) by the field staff, and a resurvey of 5% study sample to assess reliability. Anthropometric equipment was calibrated at the start of each clinic session.

## Statistical analyses

The analyses were restricted to sibling pairs of the same sex to minimise the potential for bias from difference in associations between sexes. As levels of missing data were low, only participants with complete data were analysed (Table 1). The main outcome was difference in body fat between siblings. Distributions of all variables were visually inspected, and any major outliers removed. Simple linear regression was used to describe the unadjusted relationship

**Table 1. Distribution of diet, activity, and obesity in 1,232 same-sex rural-urban migrant sibling pairs (N = 2,464 participants) of the Indian Migration Study, 2005 to 2007.**

| Participant characteristics | | % Missing values | Median (LQ, UQ), % | | Median absolute difference within sibling pairs | Median percentage difference within sibling pairs |
|---|---|---|---|---|---|---|
| | | | Rural sibling | Urban sibling | | |
| Percentage body fat (%) | | 1.6% | 23.2 (17.2, 29.1) | 29.1 (24.1, 33.5) | 5.0 | +18% |
| BMI (kg/m$^2$) | | 0.1% | 21.4 (18.8, 24.3) | 24.5 (22.2, 26.8) | 2.7 | +12% |
| Age (years) | | 0% | 41 (32, 49) | 44 (36, 50) | 3 | +7% |
| Gender | Male | 0% | 69.2% | | - | - |
| | Female | | 30.8% | | - | - |
| SLI (scale max. 38) | | 0% | 14 (10, 20) | 24 (21, 26) | 8 | +36% |
| Time lived in urban area (years) | | 0.1% | - | 24 (14, 30) | - | - |
| Factory site | Lucknow | 0% | - | 26.5% | - | - |
| | Nagpur | | - | 23.3% | - | - |
| | Hyderabad | | - | 31.3% | - | - |
| | Bangalore | | - | 18.8% | - | - |
| Total energy intake (calories/day) | | 0% | 2,529 (1,950, 3,256) | 2,916 (2,342, 3,614) | 360 | +12% |
| Cereal and legume intake (grams/day) | | 0% | 421 (316, 560) | 454 (355, 576) | 26 | +6% |
| Meat, fish, and poultry intake (grams/day) | | 0% | 10 (0, 33) | 16 (0, 42) | 0 | 0% |
| Dairy intake (grams/day) | | 0% | 285 (166, 436) | 328 (213, 452) | 27 | +10% |
| Fruit and vegetable intake (grams/day) | | 0% | 307 (213, 452) | 449 (326, 612) | 128 | +30% |
| Sugary food and sweets intake (grams/day) | | 0% | 31 (19, 45) | 33 (22, 47) | 2 | +7% |
| Fats and oils intake (grams/day) | | 0% | 40 (27, 55) | 47 (35, 68) | 10 | +21% |
| Alcoholic beverages intake (grams/day) | | 0% | 5 (0, 14) | 7 (0, 14) | 0 | 0% |
| Total physical activity energy expenditure (kJ/kg/day) | | 1.1% | 69 (53, 85) | 57 (49, 67) | −11 | −18% |
| Total time spent sedentary (min/day) | | 1.1% | 850 (733, 980) | 925 (830, 1030) | 71 | +8% |
| Total time spent engaged in moderate or vigorous activity (min/day) | | 0% | 199 (103, 315) | 120 (60, 186) | −69 | −60% |
| Time spent walking for leisure (min/day) | | 0% | 0 (0, 15) | 0 (0, 30) | 0 | 0% |
| Time spent watching television (min/day) | | 0% | 54 (0, 110) | 90 (60, 120) | 30 | +50% |

BMI, body mass index; LQ, lower quartile; SLI, Standard of Living Index; UQ, upper quartile

between difference in body fat (outcome) and difference in energy intake and energy expenditure on physical activity (exposures). Subsequently, multiple linear regression models were fitted, adjusting for age, sex, factory site, and years lived in an urban area to account for potential confounding by these variables. To address the primary aim, we first fitted a model including differences in energy intake and energy expenditure (exposures) and the confounders mentioned above. To address the secondary aim about role of specific behavioural changes, we

fitted a model including the specific food groups and subtypes of physical activity for which data were available, as well as confounders. All models were prespecified. In additional analyses, we included SLI in the primary models to explore the relative contribution of improvement in personal social conditions and urban environment. To estimate the proportion of difference in adiposity between rural and urban siblings attributable to differences in diet and activity, we used the same multivariable models to calculate the predicted difference in adiposity, assuming no difference in diet or activity, respectively, and subtracted this from the observed difference in adiposity. We expressed this as a percentage of the observed difference. We formally tested for interactions by sex or study centre, which were specified a priori.

## Results

Factory workers and their spouses ($N$ = 15,596) were contacted through their employee records and assessed for study eligibility (i.e., urban migrant with a rural dwelling sibling or a 25% random sample of urban nonmigrants; S1 Fig). A total of 13,695 (88%) completed the eligibility questionnaire, of which 7,594 (55%) met the eligibility criteria and were invited to take part in the clinical examination with their sibling. A total of 7,067 participants (3,525 sibling pairs and 17 without sibling) took part in the clinical examination. Of the 2,108 urban migrants with a rural sibling, 1,232 urban migrants had a rural sibling of the same sex ($N$ = 2,464) whose data were analysed. Those who took part in the clinical examination were broadly similar to the rest (S1 Table).

The characteristics of the study sample are presented in Table 1. Two-thirds of sibling pairs were male. Urban siblings were slightly older (approximately 3 years) but had a similar level of education as the rural sibling. Urban migrants had a much higher standard of living as expected, because most urban migration in India is for economic reasons. Most of the urban siblings had lived in an urban area for a considerable period (approximately 24 years). Urban siblings had higher percentage body fat (29% versus 23%) and BMI (25 versus 21 kg/m$^2$) than their rural counterpart. Total energy intake of urban siblings was 12% (360 calories/day) higher than their rural sibling. Cereals and legumes, dairy, fruit and vegetables, sugary foods, and fats and oils intake were all higher in urban migrants; however, there was a negligible difference in meat, poultry, and fish intake or alcohol intake. The urban siblings spent 18% (11 kJ/kg/day) less energy on physical activity than their rural counterparts. They spent more time being sedentary and watching television and considerably less time engaged in moderate or vigorous activity. Time spent walking for leisure and alcohol intake were not taken forward into the final regression models because of irregular distributions of these variables (a large number of respondents reported zero values).

Energy intake and energy expenditure were both strongly associated with difference in body fat between migrant and rural siblings (Fig 1A, Fig 1B). This remained largely unchanged on adjustment for age, sex, factory site, and time spent in an urban area (Table 2). The difference in energy intake and energy expenditure on physical activity was estimated to account for 4% and 6.5% of the difference in adiposity between siblings, respectively. In analyses of specific behaviours, difference in fat consumption ($p$ = 0.064), time spent in moderate or vigorous intensity activity ($p$ = 0.023), and watching television ($p$ = 0.006) were clearly associated with difference in adiposity between siblings, but differences in other diet and activity variables were not (Table 3). Adjustment for SLI markedly attenuated the associations, and only a weak association between physical activity energy expenditure and body fat remained (S2 Table, S3 Table). There was no evidence for interaction by study centre or sex (all $p$ > 0.1). The associations between men and women were similar, although we had limited power to discern sex-specific differences (S4 Table. S5 Table).

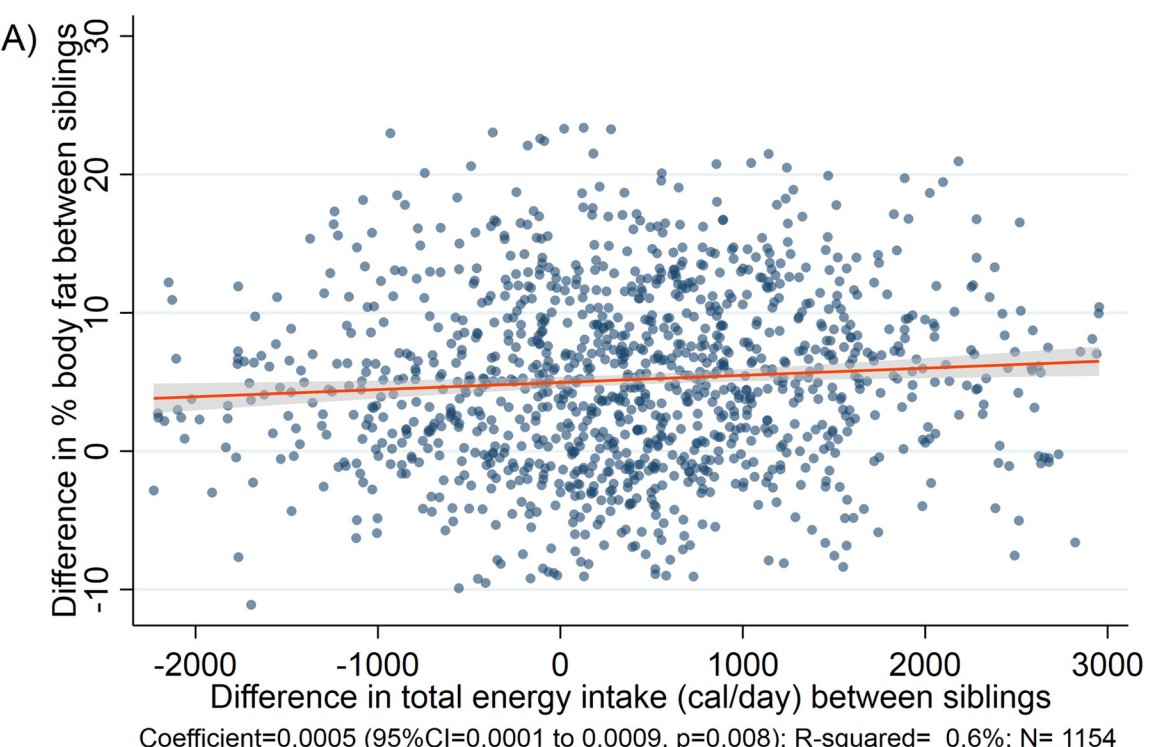

Coefficient=0.0005 (95%CI=0.0001 to 0.0009, p=0.008); R-squared= 0.6%; N= 1154

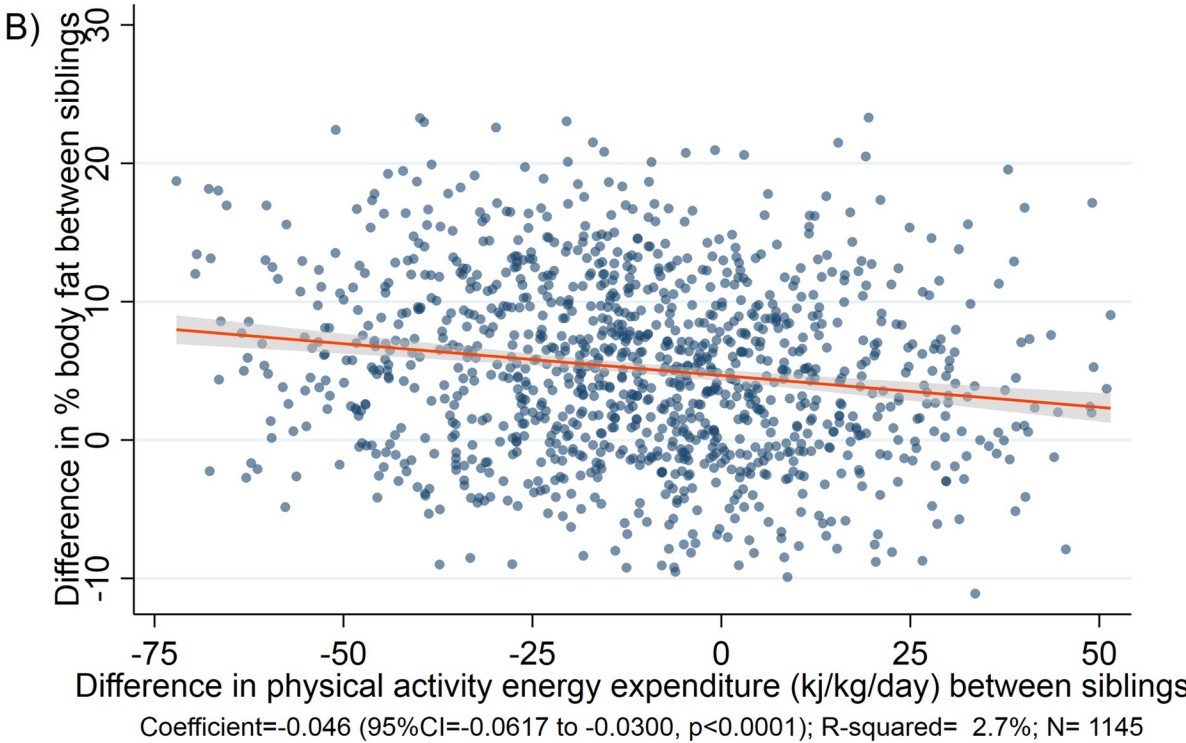

Coefficient=-0.046 (95%CI=-0.0617 to -0.0300, p<0.0001); R-squared= 2.7%; N= 1145

**Fig 1. Unadjusted linear regression models for the association of difference in energy intake or expenditure with % body fat between urban migrant and rural sibling, Indian Migration Study, 2005–2007.** A) Association of difference in energy intake and % body fat between urban migrant and rural sibling. (B) Association of difference in energy expenditure and % body fat between urban migrant and rural sibling.

**Table 2. Multivariable model for association of difference in energy intake and energy expenditure on physical activity on the difference in % body fat between urban and rural siblings in the Indian Migration Study, 2005–2007.**

| Variable | | β | 95% CI | p-value |
|---|---|---|---|---|
| Energy intake (calories/day) | | 0.001 | (0.000 to 0.001) | 0.004 |
| Physical activity energy expenditure (kJ/kg/day) | | −0.031 | (−0.046 to −0.016) | <0.001 |
| Age (years) | | 0.216 | (0.175 to 0.258) | <0.001 |
| Sex (female) | | −1.526 | (−2.329 to −0.724) | <0.001 |
| Years lived in urban area (per year) | | 0.045 | (−0.002 to 0.091) | 0.059 |
| Factory site | Lucknow | Ref. | - | - |
| | Nagpur | −1.636 | (−2.890 to −0.382) | 0.011 |
| | Hyderabad | −0.436 | (−1.409 to 0.537) | 0.380 |
| | Bangalore | −1.102 | (−2.256 to 0.052) | 0.061 |

N = 2,216 (1,108 pairs). Participants with complete data only.

β is beta-coefficient; CI is confidence interval.

Variables in the table are mutually adjusted for each other, and rural sibling are used as the reference.

## Discussion

In this study conducted in a sample of urban migrants from north, centre, and south of India and their rural dwelling siblings, greater energy intake and reduced energy expenditure both contributed to the greater adiposity of urban migrants. In terms of specific behaviours, dietary fat/oil intake, time spent on watching television, and undertaking moderate or vigorous intensity physical activity were associated with the greater adiposity among urban migrants.

**Table 3. Multivariable model for association of difference in a selected diet and physical activity behaviours on the difference in % body fat between urban and rural siblings in the Indian Migration Study, 2005–2007.**

| Variable | | β | 95% CI | p-value |
|---|---|---|---|---|
| Cereal and legume intake (grams/day) | | 0.000 | (−0.003 to 0.002) | 0.662 |
| Meat, fish, and poultry intake (grams/day) | | 0.006 | (−0.004 to 0.016) | 0.252 |
| Dairy intake (grams/day) | | 0.002 | (−0.000 to 0.003) | 0.104 |
| Fruit and vegetable intake (grams/day) | | −0.001 | (−0.003 to 0.001) | 0.544 |
| Sugary food and sweets intake (grams/day) | | −0.003 | (−0.021 to 0.016) | 0.781 |
| Fats and oils intake (grams/day) | | 0.020 | (−0.001 to 0.041) | 0.064 |
| Time spent sedentary (min/day) | | 0.000 | (−0.002 to 0.003) | 0.669 |
| Time spent in moderate or vigorous activity (min/day) | | −0.003 | (−0.006 to −0.000) | 0.023 |
| Time spent watching television (min/day) | | 0.006 | (0.002 to 0.010) | 0.006 |
| Age (year) | | 0.218 | (0.178 to 0.259) | <0.001 |
| Sex (female) | | −1.590 | (−2.390 to −0.790) | <0.001 |
| Years lived in urban area (per year) | | 0.039 | (−0.007 to 0.086) | 0.095 |
| Factory site | Lucknow | Ref. | - | - |
| | Nagpur | −2.232 | (−3.638 to −0.825) | 0.002 |
| | Hyderabad | −1.055 | (−2.153 to 0.043) | 0.06 |
| | Bangalore | −1.511 | (−2.762 to −0.260) | 0.018 |

N = 2,282 (1,141 pairs). Participants with complete data only.

β is beta-coefficient; CI is confidence interval.

Variables in the table are mutually adjusted for each other, and the rural sibling is used as the reference.

## Comparisons with previous research

Evidence on contribution of changes in diet and physical activity to increased risk of adiposity among urban migrants in India and other LMICs is lacking. Comparisons of diet and activity between urban migrants and rural populations from the area of birth of migrants points to a decline in physical activity as the primary contributor to adiposity, although the relative contribution of diet and physical activity to greater adiposity in urban migrants was not directly examined. In a study from southwestern China, urban migrants had lower energy intake than their rural counterparts, despite greater adiposity [11]. This could be attributed to higher energy requirement of rural participants who had more physically demanding occupations, but energy expenditure was not assessed in the study. Urban migrants also had greater intake of dietary fat. In a study from Peru, physical activity was found to be lower among urban migrants compared with their rural counterparts, but energy intake was not reported [12]. In a study from Guatemala, urban migrants had lower energy intake and lower physical activity; dietary variables were not associated with adiposity, but total physical activity was inversely associated with adiposity in men only [13]. In an analysis of data from six countries, including India (WHO's Study on global AGEing and adult health), occupational physical activity was lower, whereas active travel time and recreational physical activity were higher in urban migrants compared with the unrelated population living in rural areas; however, these differences were not significant in the Indian subsample, and the effects of these behaviours on excess adiposity in urban migrants was not examined [32]. Data from the National Sample Survey of India from a similar time period to our study (2004–2005) suggested no difference in energy intake between urban (2,021 calories/day) and rural (2,047 calories/day) areas of India, but higher intake of fat in urban (47 grams/day) as compared to rural areas (35 grams/day) [33]. In a standardised cross-sectional study conducted in six sites across India (2003–2005), the proportion of urban residents categorised as moderate or vigorously active (based on occupational and leisure activity) was significantly lower compared with rural residents in the same states of India [34].

## Strengths and limitations

Longitudinal studies with prospectively collected premigration data are rare because of difficulties in predicting who will migrate. Previous migrant studies have therefore relied on cross-sectional comparisons of migrant populations with host populations, or in a few cases, populations from places from which the migrants have come. However, such comparisons may be limited by unmeasured confounding from other differences in premigration characteristics of comparator populations, such as genetics, childhood home environment (including parental behaviours), and behaviours in adolescence, making attribution of observed differences to migration problematic. In this study, we used counterfactual reasoning that the rural-dwelling sibling provides an adequate control for the migrant sibling, thereby dissecting out the effect of migration from the general secular drift in environmental exposures and changes in health behaviours affecting both urban and rural populations [17]. Despite this, bias from differences in premigration characteristics of migrants and nonmigrants cannot be ruled out [9]. Migrant siblings may be chosen and supported by the family for their ability to thrive after migration and send support back to the rural family. As a result, their childhood experiences may be different despite growing up in the same home environment, resulting in important differences in lifestyle behaviours or adiposity before migration. We did not have premigration data to examine this bias, although absence of a difference in height (a global marker of selection and health, particularly in low-income settings) and educational status between migrants and nonmigrants provides some reassurance on this. The cross-sectional study design also limited the

ability to investigate the effects of short, medium, and long-term migration changes in lifestyle on adiposity, particularly because most migrants had migrated over a decade earlier.

Our response rates were lower than anticipated because of the complexity of the sibling pair design. In a majority of cases, the logistics involved at least a day to travel to the study centre and day to travel back for the rural sibling; in extreme cases, up to 3 days of travelling each way was involved. This could have introduced bias if the decision to participate was influenced by existing illness of the migrant or rural sibling. In previous analyses, we found that self-reported health of participating factory workers was similar to those who did not take part in the study (S2 Table), whereas levels of cardiovascular risk factors among rural siblings were comparable to other recent rural surveys from India [18,34].

Measurement of diet and activity by questionnaires is prone to measurement error. We used instruments that had been specifically adapted for use in this mixed rural-urban population (e.g., accounting for variations in recipes and asking open-ended questions to allow for differences in nature of activities) and found to be satisfactory on evaluation (against multiple 24-h diet recalls and accelerometery). We examined migrant and rural siblings at the same time; still, systematic recall bias by migration status cannot be ruled out [19,23].

The urban migrants were sampled from four factories in north, centre, and south of India (with rural siblings residing anywhere in the country) to take account of differences in life-styles and genetics between regions and make findings more generalisable. However, the migrant participants were all employed in factories and potentially had a better standard of living and health than the Indian population in general or the rural siblings. The study was carried out over a decade ago. The present analysis was not conducted earlier because this was not one of the original hypotheses. The pace of urbanisation and globalisation has increased dramatically over the past decade, and the profound changes in lifestyles and environment could limit the applicability of findings to contemporary populations. There is a great deal of variation in economic development across India (e.g., the gross domestic product of Delhi is eight times that of the state of Bihar); as a result, the study findings could apply to other areas of India that are at a similar stage of economic development as the study sites at that time. Furthermore, the within-sibling differences in these risk factors may still be generalisable if the societal changes affected urban and rural areas to a similar extent. It is possible that a few of the over 1,000 study villages would now be classed as urban, but this would be unlikely to have a major impact on the study findings. Nevertheless, the study findings need to be confirmed in more contemporary populations from multiple settings in India.

## Implications and further research

In common with many other LMICs, India is experiencing a rapid rise in the prevalence of obesity, particularly in its urban areas, while the population of urban areas is also increasing [4,7]. Rural to urban migrants contribute substantially to urban population growth and are particularly vulnerable to the obesogenic effects of the urban environment, highlighting them as one of the key targets for obesity interventions [6]. The findings from this study suggest that obesity interventions need to be multicomponent and address both diet and physical activity changes associated with urban living and particularly focus on behaviours such as dietary fat and oil intake, moderate to vigorous intensity physical activity, and television viewing. An important question for policy makers is the role of physical built environment [35]. In our exploratory analyses, there was a weak suggestion that the urban environment may contribute to increased risk of obesity through a decline in physical activity, independent of improvement in personal social conditions. These results should be interpreted with caution because of the challenges in measuring social conditions comparably across urban and rural areas of India

and the low statistical power of our study to investigate these differences (vast majority of urban migrants had a higher standard of living index compared with their rural siblings). Nevertheless, these findings are intriguing in the context of globalisation and changing social conditions in both urban and rural areas and warrant further research. Policy interventions in India have so far focussed on limited aspects of dietary intake (e.g., advertising and labelling of junk food); we suggest that these should be extended and expanded to include interventions aimed at increasing physical activity in urban areas, such as improving walkability of streets and green spaces, although there could be challenges to such approaches because of increasing pollution levels or rapid urbanisation of areas causing a decrease in green spaces [36]. Migrants, particularly in the workplace, are a readily identifiable group, who might be more motivated to take part in on-site health promotion activities, including healthy canteen lunches and exercise classes [7,37]. Given the variations in economic transition and lifestyles across India and other LMICs and the ongoing urbanisation and globalisation, further research is needed to confirm the findings from this study in more contemporary populations in different regions of India and other LMICs. In addition, culturally appropriate multicomponent behavioural interventions need to be robustly evaluated for effectiveness and scalability.

## Conclusions

In this study, we found that increased energy intake and reduced energy expenditure contributed equally to greater adiposity among urban migrants from four cities in north, centre, and south of India. Policies aimed at controlling the rising prevalence of obesity in urbanising India need to be multicomponent and target both energy intake and expenditure, with a specific focus on behaviours such as dietary fat and oil intake, time spent on watching television, and time spent engaged in moderate or vigorous intensity physical activity. Similar research is needed from more contemporary populations and other regions of India and LMICs to confirm these findings and develop context-specific interventions.

## Supporting information

**S1 STROBE checklist.**
(DOCX)

**S1 Text. Indian Migration Study questionnaire.**
(DOC)

**S1 Fig. Recruitment flow chart, Indian Migration Study, 2005–2007.**
(DOC)

**S1 Table. Participant characteristics by response, Indian Migration Study, 2005–2007.**
(DOC)

**S2 Table. Multivariable model for energy intake and energy expenditure, adjusted for Standard of Living Index.**
(DOCX)

**S3 Table. Multivariable model for selected diet and physical activity behaviours, adjusted for Standard of Living Index.**
(DOCX)

**S4 Table. Multivariable model for energy intake and energy expenditure, stratified by sex.**
(DOCX)

**S5 Table. Multivariable model for selected diet and physical activity behaviours, stratified by sex.**
(DOCX)

## Acknowledgments

We wish to acknowledge the investigators, field teams, and participants of the Indian Migration Study.

## Author Contributions

**Conceptualization:** Sanjay Kinra, Kolli Srinath Reddy, George Davey Smith, Yoav Ben-Shlomo, Shah Ebrahim.

**Formal analysis:** Poppy Alice Carson Mallinson, Jenny A. Cresswell, Liza J. Bowen.

**Funding acquisition:** Sanjay Kinra.

**Investigation:** Sanjay Kinra, Dorairaj Prabhakaran, Kolli Srinath Reddy, Anura V. Kurpad, Yoav Ben-Shlomo, Shah Ebrahim.

**Methodology:** Sanjay Kinra, Jenny A. Cresswell, Tanica Lyngdoh, Kolli Srinath Reddy, Mario Vaz, Anura V. Kurpad, Yoav Ben-Shlomo, Shah Ebrahim.

**Project administration:** Sanjay Kinra, Tanica Lyngdoh.

**Resources:** Sanjay Kinra.

**Writing – original draft:** Sanjay Kinra.

**Writing – review & editing:** Sanjay Kinra, Poppy Alice Carson Mallinson, Jenny A. Cresswell, Liza J. Bowen, Tanica Lyngdoh, Dorairaj Prabhakaran, Kolli Srinath Reddy, Mario Vaz, Anura V. Kurpad, George Davey Smith, Yoav Ben-Shlomo, Shah Ebrahim.

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
