## [Editor Report · Decision Letter 0]

17 Feb 2020

Dear Dr Kinra, 

Thank you for submitting your manuscript entitled "Role of diet and physical activity in increased risk of obesity among rural to urban migrants in India: the Indian Migration Study" for consideration by PLOS Medicine.

Your manuscript has now been evaluated by the PLOS Medicine editorial staff and I am writing to let you know that we would like to send your submission out for external peer review.

Kind regards,

Helen Howard, for Clare Stone PhD 

Acting Editor-in-Chief

PLOS Medicine 

plosmedicine.org

---

## [Decision Letter · Decision Letter 1]

23 Mar 2020

Dear Dr. Kinra,

Thank you very much for submitting your manuscript "Role of diet and physical activity in increased risk of obesity among rural to urban migrants in India: the Indian Migration Study" (PMEDICINE-D-20-00490R1) for consideration at PLOS Medicine. 

[LINK]

In light of these reviews, I am afraid that we will not be able to accept the manuscript for publication in the journal in its current form, but we would like to consider a revised version that addresses the reviewers' and editors' comments. Obviously we cannot make any decision about publication until we have seen the revised manuscript and your response, and we plan to seek re-review by one or more of the reviewers. 

We expect to receive your revised manuscript by Apr 13 2020 11:59PM. Please email us (plosmedicine@plos.org) if you have any questions or concerns.

We look forward to receiving your revised manuscript. 

Sincerely,

Adya Misra, PhD

Senior Editor 

PLOS Medicine

plosmedicine.org

Please revise your title according to PLOS Medicine's style. Your title must be nondeclarative and not a question. It should begin with main concept if possible. "Effect of" should be used only if causality can be inferred, i.e., for an RCT. Please place the study design ("A randomized controlled trial," "A retrospective study," "A modelling study," etc.) in the subtitle (ie, after a colon).

Abstract

Abstract Background: Provide the context of why the study is important. The final sentence should clearly state the study question.

Please combine methods and findings into one section. The last sentence of this section should be a limitation of your methodology

Please briefly mention the previous study in the background

In the methods and findings section please mention when the study was carried out

Please include a conclusions section, summarising the study findings. Please address the study implications without overreaching what can be concluded from the data; the phrase "In this study, we observed ..." may be useful.

* Please interpret the study based on the results presented in the abstract, emphasizing what is new without overstating your conclusions.

* Please avoid vague statements such as "these results have major implications for policy/clinical care". Mention only specific implications substantiated by the results.

* Please avoid assertions of primacy ("We report for the first time....") Please remove the interpretation subsection. 

Introduction

References- please place a full stop after the square brackets. For example : rural to urban migration [7,8].

“…notably childhood home

environment (including parental behaviours) and behaviours, and to a lesser extent

genetics”. I think this needs to be clarified as genetics do play a strong role in adiposity and there is emerging evidence to suggest that it plays a role in diet/physical activity too

Please provide the dates of the Indian Migration study here 

Please clarify this sentence “Based on data from elsewhere, we hypothesised that the greater adiposity in urban migrants was primarily

due to the differences in their physical activity”

Please clearly state the hypotheses being tested in this study and outline the aims. 

Please clarify why the study results were not published sooner-since migration, diet and behaviours have changed since the original study was carried out. This information should be provided in the methods and or discussion sections. 

Methods

Please clarify if the nutrient databases are available online and provide a citation 

Please provide a list of the open ended questions used to determine physical activities specific to participants

Please cite the individual guidelines used to determine diet and physical activity levels in participants

Please expand “appropriate adjustments were made using standard methods” citing the methods used

Has the definition of “urban” changed since this study was completed? Please mention this in the discussion section as a potential limitation. Same goes for any potential recall bias in diet and physical activity measurements

Please clarify the time points at which skin fold thickness was measured

Please provide the name(s) of the institutional review board(s) that provided ethical approval.

Please report the study according to STROBE guidelines, providing a completed STROBE checklist as supplementary information. Please mention at the beginning of the methods section that the study has been reported in accordance with STROBE guideline and the checklist is provided as SI file xx

Please present and organize the Discussion as follows: a short, clear summary of the article's findings; what the study adds to existing research and where and why the results may differ from previous research; strengths and limitations of the study; implications and next steps for research, clinical practice, and/or public policy; one-paragraph conclusion. 

“increasing physical activity in urban areas (e.g. improving walkability of streets and

green spaces)” could there be challenges to this? Increasing pollution levels or the rapid urbanisation of areas causing a decrease in green spaces? Please discuss 

Comments from the reviewers:

Reviewer #1: I confine my remarks to statistical aspects of this paper. Unfortunately, I think there are some problems with the analysis

The authors need to add standard of living to the regression. Urban dwellers had much higher standard of living and this is surely related to diet (and probably to exercise and sedentary time). 

In the figures, it is clear that there are high leverage points (which inflate R^2) and outliers (which can change the coefficients. 

I would not change all the variables into percents. I would leave the DV as "difference in % body fat" (not divide by the urban dweller) and leave the IVs as differences in grams per day (or minutes per day) rather than %s. This would be easier to interpret.

How were variables selected for the multivariable models? Why are there different variables in the two models?

NOTE: Line numbers would have made the review easier.

p. 2 The section about fruits, vegetables etc at the end of results needs to be reworded to include "significant". 

p. 4 Please give numbers for reliability rather than just saying "acceptable"

p. 5 Same comment - give numbers for reliability and validity

 BMI - why use BMI when you have the more accurate skinfold measurements? 

p. 7 Please comment on the huge difference in standard of living. 

Peter Flom

Reviewer #2: Thank you for the opportunity to review the study by Kinra et al., investigating the role of diet and physical activity in increased risk of obesity among rural to urban migrants in India: the Indian Migration Study. 

The cross-sectional Indian Migration study has been undertaken between March 2005 and December 2007 and has been described in detail in the paper 'The effect of rural-to-urban migration on obesity and diabetes in India: a cross-sectional study' published by Ebrahim et al. in PLOS Medicine in April 2010. The main hypothesis of the initial paper was that rural-urban migrants would have higher rates of obesity and diabetes than rural non-migrants. The authors report migration into urban areas to be associated with increases in obesity, and amongst others report results showing physical activity to be decreased and fat intake to be increased in the rural-urban sibling migrants. They observe gender differences in some risk factors by place of origin that require further explanation. 

The current study by Kinra et al extends the previous cross-sectional analysis addressing the relative contribution of changes in diet and physical activity to risk of obesity in urban migrants. Major results of the paper, i.e. 11% higher energy intake (mainly through fat intake) and 17% lower energy expenditure by physical activity accounting for 2.3% and 6.9% of the difference in adiposity between siblings, add to the results described by Ebrahim and colleagues, yet provide limited additional information to be used for the development of targeted prevention / intervention programmes. 

Whilst the initial paper made the case for sex-specific analysis of life-style related factors, the current analysis presents results adjusted for sex rather than sex-specific results, generally considered important for tailored prevention and intervention programmes. Further, the initial, very well conducted study, has been undertaken 13-15 years ago limiting applicability of results for prevention and intervention efforts within the context of profound life-style changes since 2005 linked to e.g. increasing urbanization and globalization of e.g. food commodities such as availability and affordability of ultra-processed food and possibly other societal factors influencing life-style factors of the general population residing both in rural and urban areas. 

Further comments: 

Given the significant difference in age between siblings residing in rural and urban India, presentation of age-adjusted graphs would be preferable to take into account any age-related differences, such as for example the well-known correlation of age with higher percentage of body fat. 

In line with the initial paper a discussion on the strengths and limitations of the sibling design as compared to other designs for the study of migration-related changes, the potential bias related to the low response rate, as well as a discussion on the ability of the current study to investigate the effect of short, medium and long-term migration on changes in risk factors and subsequent cardio-metabolic outcomes of rural-urban migrants are warranted. 

Reviewer #3: This is a well-designed, interesting and useful study. Also very well written. I have very minor comments below.

The sample is described as geographically representative? Does this mean the proportion of participants from the North, central and Southern India matched the proportion of the Indian population in each of these regions?. It would be helpful to understand why this is important- i.e.: that similar patterns were seen in all three regions so in cities throughout India rural-urban migrants are likely to make similar behavioural changes (despite the different cultures in Northern, central and Southern India)?

In the Methods section: "each participant was asked to invite one non-migrant full sibling of the same sex and closest to them in age still residing in their rural place of origin" but then later: "The analyses were restricted to sibling pairs of the same sex to minimise the potential for bias from difference in associations between sexes". And then in the first paragraph of the results there is no suggestion that eligibility required a sibling of the same sex. This is just slightly confusing, I wonder whether the methods should have read "each participant was asked to invite one non-migrant full sibling, *preferably* of the same sex and closest to them in age" (i.e. there were mixed sex pairs in the study but not in this analysis?).

Although the design of this study means that childhood circumstances broadly are similar between the two groups, and genetics also 50% shared- I think it is worth considering how the migrating sibling is often chosen and supported by a family- having the opportunity to migrate and in return sending support back to their rural family (so the family chooses the sibling most likely to thrive after migration). Urban siblings were slightly older (3 years) this suggests the possibility of other systematic differences between the migrant sibling and the rural sibling which might confound the findings. E.g.: despite the same childhood home they may have had different childhood experiences within that home including the often older, migrant sibling getting the lion's share of resources- perhaps means they have different dietary and physical activity behaviours and a different BMI even before migration and these things aren't due to exposure to the urban environment (particularly noting that years in the urban environment was not significant in the model). Agreed that the lack of difference in height and education is reassuring. I think it would still be worth stating this in the limitations. 

In the discussion: "existing evidence on behavioural changes associated with rural to urban migration in LMICs is limited and points to a decline in physical activity as the primary change" also " there are no comparable studies of rural to urban migration from India". In analysis of the WHO-SAGE survey which includes data from India (https://journals.plos.org/plosone/article?id=10.1371/journal.pone.0122747 ) we found the rural to urban migrants achieved significantly less occupational physical activity although more leisure time physical activity and active travel. Although the rural to urban migrants were not siblings of the rural participants so may have different by home environment and genetics, so fair enough to say this was not comparable- but still relevant to this discussion (although I know cheeky to be referring to a paper of which I'm an author).

[LINK]

---

## [Decision Letter · Decision Letter 2]

10 Jun 2020

Dear Dr. Kinra,

Thank you very much for re-submitting your manuscript "Relative contribution of diet and physical activity to increased adiposity among rural to urban migrants in India: A cross-sectional study" (PMEDICINE-D-20-00490R2) for review by PLOS Medicine.

I have discussed the paper with my colleagues and the academic editor and it was also seen again by reviewers. I am pleased to say that provided the remaining editorial and production issues are dealt with we are planning to accept the paper for publication in the journal.

[LINK]

We look forward to receiving the revised manuscript by Jun 17 2020 11:59PM. 

Sincerely,

Adya Misra, PhD

Senior Editor 

PLOS Medicine

plosmedicine.org

Requests from Editors:

I note the questionnaire is cited, but if possible please include here as a Supp file.

Please ensure the STROBE uses sections and paragraph numbers instead of lines as these wont be in the final version. also page numbers change on formatting, so please avoid those too. 

Age of data - it is noted that the data are old. can this be updated or is it possible to comment on in the MS? 

Comments from Reviewers:

Reviewer #1: The authors have addressed my concerns and I now recommend publication

Peter Flom

Reviewer #2: Thank you for responding in detail to the initial review comments. Yet, I do still doubt whether the current paper adds significantly to the original paper from Shah Ebrahim et al who already addressed the relative contribution of changes in diet and physical activity to risk of obesity in urban migrants.

Further, I consider sex-specific analysis of major importance as in most contexts, including in India, dietary behaviour and physical activity patterns are gendered, so that information specific to women, respectively men for the purpose of the development and implementation of tailored prevention and intervention programmes is warranted. 

Reviewer #3: The authors have addressed all my comments appropriately and I am happy with this draft of the paper.

Thanks also for the opportunity to read the comments from the editor and other two reviewers. I think the authors have also done a good job of responding to these points.

In terms of reviewer 1 "The authors need to add standard of living to the regression. Urban dwellers had much higher standard of living and this is surely related to diet (and probably to exercise and sedentary time)." Just to say, I agree with the authors, that adding standard of living to the regression would examine a different question than the one they have asked. Having read reviewer 1's comments though, I do wonder whether it is worth it (in addition to the results they have presented) because it would allow them to infer what the contribution of *urbanisation* rather than *increased standard of living*- i.e.: if rural populations had a similar increase of wealth would we expect similar changes in diet, physical activity and hence obesity or not? I.e.: What is the contribution of the *urban environment* independent of the *standard of living* changes. It might be particularly interesting if the association between energy intake or energy expenditure and obesity, but not both, were extinguished.

[LINK]

---

## [Editor Report · Decision Letter 3]

8 Jul 2020

Dear Prof. Kinra, 

On behalf of my colleagues and the academic editor, Dr. Oyinlola Oyebode, I am delighted to inform you that your manuscript entitled "Relative contribution of diet and physical activity to increased adiposity among rural to urban migrants in India: A cross-sectional study" (PMEDICINE-D-20-00490R3) has been accepted for publication in PLOS Medicine. 

PRODUCTION PROCESS

PRESS

PROFILE INFORMATION

Thank you again for submitting the manuscript to PLOS Medicine. We look forward to publishing it. 

Best wishes, 

Clare Stone, PhD

Managing Editor 

PLOS Medicine

plosmedicine.org